# Experimental Investigation on Seepage Characteristics of Clay–Structure Interface after Shear Deformation

**DOI:** 10.3390/ma15113802

**Published:** 2022-05-26

**Authors:** Jiacheng Tan, Zhenzhong Shen, Liqun Xu, Hongwei Zhang, Yingming He

**Affiliations:** 1State Key Laboratory of Hydrology-Water Resources and Hydraulic Engineering, Hohai University, Nanjing 210098, China; tjccccc@hhu.edu.cn (J.T.); zhzhshen@hhu.edu.cn (Z.S.); 2College of Water Conservancy and Hydropower Engineering, Hohai University, Nanjing 210098, China; 1902010125@hhu.edu.cn

**Keywords:** clay–structure interaction, mechanical properties, seepage characteristics, shear dilation, interface roughness

## Abstract

There exist shear and seepage behaviors on the interface between clay core and concrete slab in clay core dams. In order to investigate the seepage characteristics of the clay–structure interface after shear deformation, a shear-seepage test system is proposed, in which the seepage direction is perpendicular to the shear direction. The shear test and shear-seepage test are performed on clay–metal and clay–mortar interfaces under different normal stresses (100, 200, 400, 800, and 1600 kPa). The shear stress-deformation curves of two clay–structure interfaces exhibit softening behavior and residual friction behavior. The interface roughness can enhance peak and residual shear strength and increase peak displacement. The shear-seepage test results show that specimen permeability decreases first and then increases to a stable value as shear deformation increases under low normal stress, while it decreases continuously and then retains stability under high normal stress. The interface roughness enhances specimen permeability under low normal stress, whereas it has a weak effect on specimen permeability under high normal stress. Compared with initial permeability, shear deformation reduces specimen permeability rather than raise it. The ratio of stabilized permeability coefficient to initial value ranges from 0.6 to 0.8. The clay–structure interface still has a good resistance to seepage failure after shear deformation. The shear dilation features and interface pore decrease caused by shear behavior are the internal attributions of clay–structure specimen permeability evolution.

## 1. Introduction

Clay core wall is an important anti-seepage structure, which is widely used in rockfill dams, and is often built on concrete slabs. In the case of steep bank slopes, large shear deformation may occur at the interface between the clay core and structure. So, a high plasticity clay layer will be set at the bottom of the clay core to adapt to large shear deformation in actual engineering projects, such as Lianghekou (height of 295 m), Nuozhadu (height of 261.5 m), and other high clay–core rockfill dams [1,2,3,4]. After the reservoir is impounded, the interface between clay and structure will be subjected to shear deformation and seepage, so that the stress-deformation-seepage state of the interface is usually complex. The seepage direction is parallel to the water flow direction; while the shear deformation direction is along the slope direction. The seepage direction is perpendicular to the shear direction, which is depicted in Figure 1. Consequently, seepage channels can be easily developed through the weak clay–structure interface due to the difference in mechanical properties between the clay and structure [5,6,7].

For instance, the Teton dam, due to the uneven deformation of clay on the bank slope, resulting in seepage failure at the interface between clay and rock, eventually collapsed [8,9]. The Fontenelle dam in the US, a core wall earth dam, collapsed in 1965. After the accident investigation, the main reason for dam failure was attributed to the internal erosion at the interface between clay and concrete [10,11]. Therefore, it is important to investigate the effects of shear deformation on seepage characteristics of clay–structure interface.

The shear-seepage problem of clay–structure interface is a hydro-mechanical coupling problem involving stress, deformation, and seepage stability. However, most of the previous studies only consider one single factor, mainly including three aspects. The first one is focused on the mechanical properties of clay–structure interface [12,13,14,15,16], ignoring the hydraulic effect. The second one is related to clay–structure seepage characteristics [17,18,19,20] without considering the influence of stress and deformation. The third aspect is to investigate the shear-seepage behavior of clay soil by carrying out the compression-shear coupling penetration tests [21,22,23] and the triaxial penetration tests [24,25,26], but these studies do not involve soil-structure interface.

Generally, there has been very limited study on seepage characteristics of clay–structure interface after shear deformation. Lei et al. [27] and Liu et al. [28] primarily carried out the shear-seepage test under the condition of vertical shearing and seepage with a modified triaxial instrument. The effects of normal stress, shear deformation, and hydraulic gradient on seepage characteristics of clay–structure interface were analyzed. However, the directions of shear deformation and seepage are parallel in their tests, and it is inconsistent with the actual problem. Luo et al. [29,30] and Wang et al. [31,32] also used a novel soil-structure contact erosion apparatus to simulate the stress-seepage state at the joint of clay–core wall and concrete cut-off wall and investigated the hydro-mechanical coupling characteristics of the joint. Deng et al. [33] and Zhang et al. [34] developed a set of test equipment for impermeability characteristics of junctional zone between soil and hard surface and implemented the experimental studies on a compacted natural broadly graded clayey soil. The permeability evolution mechanism during shear deformation is primarily explained by fabric adjustment. The existing research has confirmed that normal stress and shear deformation have an important influence on the seepage characteristics of clay–structure interface.

However, due to the limitation of test facilities, there are still some experimental problems unsolved, such as large shear deformation cannot be applied, the relationship between the shear direction and the seepage direction is not clear, and the specimen preparation process and test method are not perfect. In addition, interface roughness, as an important factor of soil-structure interaction, has been widely studied on interface mechanical properties [35], but the effect of interface roughness on permeability has not been reported yet.

In this consideration, there is still a lack of systematic understanding of the shear-seepage behavior mechanism of clay–structure interfaces under the condition of high normal stress, large shear deformation, and high hydraulic gradient. Therefore, a shear-seepage test system is proposed, and a series of shear tests and shear-seepage tests are conducted on clay–metal and clay–mortar interfaces. The characteristics of seepage perpendicular to shear direction is obtained. The influences of normal stress, shear deformation, and interface roughness on the mechanical properties and seepage characteristics are systematically investigated. This paper is helpful to better understand the shear-seepage behavior mechanism of clay–structure interface.

## 2. Materials and Methods

### 2.1. Shear-Seepage Test System

The novel shear-seepage test system shown in Figure 2 consists mainly of a confining pressure system, a rotary shear system, a seepage pressure system, a shear-seepage pressure chamber, an automatic data acquisition system, and a specimen preparation system.

The normal stress is provided by the GDS pressure-volume controller of the confining pressure system. The largest confining pressure is 3.0 MPa, and the volume capacity is 200 mL. The rotary shear system produces large shear deformation at the clay–structure interface by applying torque to the rotating shaft. The shear rate is the rotation rate, which ranges from 0.01 to 100 mm/min. The shear deformation can be infinite theoretically. The seepage pressure system comprises a seepage pressurized source, a water vapor exchange tank, a seepage pressure output control gauge, and so on. The measurement range of the seepage pressure is 0–0.8 MPa with a precision of 1.0 kPa. Combined with the specimen height (50 mm), the maximum hydraulic gradient can theoretically reach 1600. Therefore, this novel shear-seepage test system can provide high normal stress, large shear deformation, and high hydraulic gradient. The automatic data acquisition system contains a torque transducer, an angle transducer, two electronic balances, and a set of data acquisition software. The torque transducer and the angle transducer are used for measuring the shear stress and the shear displacement, respectively. Two electronic balances are used to record the seepage flow at the seepage water inlet and outlet. All data is connected to the collection box through the sensor and then transmitted to the computer for fully automatic real-time monitoring.

Figure 3 shows the designed shear-seepage pressure chamber, which is mainly composed of pressure chamber top, upper porous plate, pressure chamber barrel, clay–structure specimen, lower porous plate, and pressure chamber base.

The rubber membrane divides the pressure chamber into two sealed cavities, the outer cavity is used to store high confining pressure water, and the inner cavity is used to place the clay–structure specimen. The seepage water inlet and outlet are set into the pressure chamber base and top, respectively. In order to avoid coaxial rotation of clay specimen when the structure rod rotates, the upper and lower porous plates are equipped with eight blades (thickness of 2 mm), which are embedded in clay during specimen preparation. In addition, a set of specimen preparation system is designed to match the specimen size. The structure material is cast on a prefabricated rotating shaft, and the ring-shaped columnar soil is wrapped around the structure. The specimen preparation method can ensure that the rotating shaft, structure material, and clay soil are coaxial. In this case, the clay–structure interface is a cylindrical surface. The shear direction is the circumferential direction of clay–structure interface, and the seepage direction is along the generatrix direction of the interface. The directions of shear deformation and seepage are orthogonal, which is consistent with the actual project.

### 2.2. Testing Materials

The clay soil of a clay–core rockfill dam in southwest China was used in this study. Table 1 shows the basic physical properties of the clay soil material. The clay soil was classified as CL according to the unified soil classification system (USCS) [36]. The grain size distribution of the clay soil is shown in Figure 4, which is measured by Malvern 2000 laser particle size analyzer. Through X-ray diffraction analysis, the main component of clay is SiO_2_, accounting for 59.9%, R_2_O_3_ and K_2_O accounting for 28.1% and 4.2%, respectively.

The structure rods used in this paper are shown in Figure 5, including the metal rod (Figure 5b) and the mortar rod (Figure 5c). The metal rod is prefabricated from stainless steel. The mortar is cast on the prefabricated rotating shaft (Figure 5a) by the matching specimen mold to ensure that the mortar rod is tightly attached to the rotating shaft. According to the actual concrete mix ratio requirements, the water-cement ratio is 0.42, and the sand–cement ratio is 2.5:1. The mortar rod was cured for 28 days in standard condition. The clay–structure specimen (see Figure 5d) has an outer diameter of 61.8 mm, an inner diameter of 20 mm, and a height of 50 mm.

### 2.3. Testing Methods

Specimen preparation. Firstly, the clay soil was dried, crushed, and sieved. The clay and water were mixed according to the designed dry density of 1.83 g/cm^3^ and water content of 15.6%, then the prepared clay soil was stored for 24 h. Secondly, the prepared structure rod was placed at the center of the lower porous plate and the specimen was compacted directly on the lower porous plate to reduce the disturbance to the specimen, which could ensure that the clay soil adheres to the structure rod. The clay was evenly divided into two layers and compacted, and the target compaction degree of each layer was 98%. The joint surface of the two layers was fully shaved to facilitate the close combination of the soil. Thirdly, the upper porous plate was pressed on the soil, and the rubber membrane was wrapped on the upper porous plate, clay–structure specimen, and lower porous plate. Finally, the specimen was saturated in a vacuum saturated bucket for 24 h.Specimen consolidation. Firstly, fill all pipes with airless water and install the pressure chamber. Secondly, inject distilled water into the pressure chamber from the confining pressure inlet until it is full, and then block the vent hole. Finally, apply the specified consolidation pressure to the water in the pressure chamber by using the GDS pressure-volume controller, and open the seepage water outlet and the seepage water inlet to drain the pore water of the specimen. The volume change in this stage was measured by the variation of pore water. When high consolidation pressure was needed, a hierarchical loading method was adopted. When the volume change of the GDS pressure-volume controller was less than 1 mm^3^/min, or no pore water was discharged, the specimen consolidation was considered complete.Shear-seepage test. The shear-seepage test mainly included two processes: seepage process and shear process, which were carried out alternately. During the seepage process, open the seepage water inlet and outlet and apply the seepage pressure. The upward seepage water flowed into the specimen and discharged from the seepage water outlet. When the specimen seepage outflow was equal to outflow, the seepage flow was considered to be stable. Then, we recorded the inlet and outlet hydraulic heads and the inlet and outlet seepage discharge, and calculated the overall permeability coefficient of the specimen (defined by Equation (1)). The whole seepage process was no less than 6 h. After the current seepage process was completed, we closed the seepage pressure system and left the specimen for several hours to ensure that the residual water at the clay–structure interface was fully discharged. During the shear process, we applied torque to the rotating shaft and produce large shear deformation at the clay–structure interface. When the shear deformation reached the specified displacement, we stopped shearing. After that, we repeated the above seepage process and measured the overall permeability coefficient of specimen under different shear deformation conditions. Specimens should be changed under different normal stress. The experiment was performed in the laboratory at room temperature of approximately 20 °C. The experimental flowchart is shown in Figure 6.

### 2.4. Testing Schemes

Subjected to the normal stresses of 100, 200, 300, and 400 kPa, the conventional direct shear tests were conducted on the saturated clay. Meanwhile, rotary shear tests were conducted on the clay–structure interface under normal stresses of 100, 200, 400, 800, and 1600 kPa to analyze the mechanical properties. As specified by ASTM D5321 [37], a shear displacement rate of 1.0 mm/min was used in this study. The shear process was terminated when the tangential displacement reached 20 mm (about 30% tangential strain).

During the shear process, the stress-deformation state of the interface will change continuously as shear displacement increases, which may have different effects on specimen permeability. Therefore, based on rotary shear test results, seven shear displacement control points were selected from the shear stress-shear displacement relationship of clay–structure interface. The control points were 0, 50%, 100%, 200%, and 500% of the peak displacement value and 60% and 100% of the maximum shear displacement value. The applied seepage pressure was 40% of the normal stress to prevent seepage failure caused by excessive water pressure. The permeability test was performed on the clay–structure specimen after each shear stage. The shear-seepage test schemes are listed in Table 2.

### 2.5. Overall Permeability Coefficient

In this study, the structure rods are considered impermeable compared to clay soil. Therefore, the overall permeability coefficient of clay–structure specimen can be expressed as Equation (1) according to Darcy’s law [38].
(1)$$k=\frac{q}{\pi (r_{2}^{2}-r_{1}^{2})}\times \frac{h}{H_{1}-H_{2}}$$
where $r_{1}$ is the inner diameter of clay, which equals the radius of the structure rod. $r_{2}$ is the outer diameter of clay. h is the height of specimen. q is the stable seepage flow rate, which can be taken as the inflow or outflow rate. $H_{1}$ is the inlet head and $H_{2}$ is the outlet head.

## 3. Results

### 3.1. Mechanical Behavior Analysis

#### 3.1.1. Shear Behavior

The direct shear test results of clay soil are shown in Figure 7a, and the relationships between shear stress and displacement for clay–metal and clay–mortar interface are plotted in Figure 7b,c, respectively.

Figure 7a suggests that there exists a slight softening phenomenon under 100 and 200 kPa, and a hardening phenomenon under 300 and 400 kPa. The clay peak strength increases as the normal stress increases. For the clay–structure interface, all the curves exhibit softening behavior, especially on the clay–metal interface. The shear stress-deformation curves of the clay–structure interface have obvious peak strength and residual strength. Both the peak and residual shear strength increase with the increase of normal stress. The shear test results are consistent with Chen et al. [35] and Taha et al. [13].

Taking the clay–mortar interface under the normal stress of 1600 kPa as an example, four friction stages are identified and presented for the clay–mortar interface shear behavior (see Figure 7c). In the first stage, which is known as the linear elastic stage, the shear stress increases linearly with the shear displacement from 0 to 1.4 mm (OA section). Subsequently, the nonlinearity becomes clear and the increment rate of shear stress decreases before peak shear strength (AB section). After that, the clay–mortar interface becomes shear failure and enters the third stage, namely softening stage (BC section). In this stage, shear stress decreases quickly as shear displacement increases and the interface is further destructed. When the shear displacement reaches 7 mm (point C), the interface has been fully destroyed and the interfacial shear stress becomes stable. And then the shear process enters the final stage, which is the residual friction stage (CD section). The shear stress is basically constant during the shear process and the stable shear stress value (point D) is the residual strength of the interface. In Figure 7b, due to the interface roughness, there is no nonlinear rising stage for clay–metal interface shear behavior. Chen et al. [35] performed a series of direct shear tests on different types of red clay–concrete interfaces and found the same changing pattern of shear behavior.

In addition, the peak displacement values of clay soil are greater than those of the clay–structure interface, which suggests that clay soil has better ductility. The peak displacement values of the clay–mortar interface are 0.4, 0.5, 0.8, 1.2, 1.8 mm and those of the clay–metal interface are 0.3, 0.5, 0.4, 0.5, 0.8 mm for normal stresses of 100, 200, 400, 800, and 1600 kPa, respectively. The clay–mortar interface has larger peak displacement values under the same normal stress, which indicates that the interface roughness increases peak displacement and improves the shear ductility. In addition, the peak displacement increases as the normal stress increases, which suggests that the ductility of the interface is enhanced by an increase in the normal stress. The hindering effect of normal stress is a reason for the specimens’ ductility improvement under higher normal stress.

#### 3.1.2. Shear Strength

For a better understanding of the shear strength of clay soil and clay–structure interface, relationships between normal stress and shear stress (peak and residual) are plotted in Figure 8.

Both the peak and residual shear stress–normal stress curves can be well fitted by the linear Mohr–Coulomb failure law:(2)$$\tau_{p}=c_{p}+\sigma \tan\varphi_{p}$$
(3)$$\tau_{r}=c_{r}+\sigma \tan\varphi_{r}$$
where $\tau_{p}$ and $\tau_{r}$ are peak and residual shear strength, respectively. σ is the normal stress. $c_{p}$ is the peak cohesive force. $c_{r}$
is the residual cohesive force. $\varphi_{p}$ is the peak internal frictional angle, and $\varphi_{r}$ is the residual internal frictional angle. The shear strength parameters for the Mohr–Coulomb failure law are regressed linearly based on the test data by using the least square method. The correlation coefficients of the fitted curves are all above 0.99. The strength parameters of clay and clay–structure interfaces are listed in Table 3.

From Figure 8 and Table 3, it is observed that the clay shear strength is greater than the clay–structure interfacial shear strength under the same normal stress. It indicates that the existence of interface reduces the shear resistance, and the shear failure plane is more likely to develop along the interface. For clay–structure interface, the strength parameters of the clay–mortar interface are larger than those of the clay–metal interface. The peak and residual shear strength values of two clay–structure interfaces are very close to each other under low normal stress. With the increase of the normal stress, the gap between them becomes larger and larger. Additionally, the gap between the peak and residual strength of the clay–mortar interface is significantly beneath that of the clay–metal interface. Because the metal interface is smoother than the mortar interface and the friction coefficient is smaller, the softening behavior is more obvious on the clay–metal interface. These findings show that the interface roughness has a remarkable effect on the interfacial shear behavior and shear strength. The higher interface roughness may be beneficial to the improvement of interfacial shear strength.

### 3.2. Seepage Behavior Analysis

#### 3.2.1. Contact Seepage of Initial Interface

Figure 9 shows the contact seepage results of the initial interface (without any shear deformation under different normal stresses).

It is very clear that overall permeability coefficients of clay soil and clay–structure specimen both decrease as the normal stress increases, and the decreasing rate becomes smaller. This suggests that the normal stress has remarkable effects on interface seepage. From the test data points, the overall permeability coefficient of clay–structure specimen is obviously higher than that of the clay specimen, especially under the normal stress of 100 kPa, which indicates that the existence of an interface increases specimen permeability. Comparing two clay–structure specimens, the overall permeability coefficient of clay–mortar specimen is larger than that of the clay–metal specimen under low normal stress, but the values are basically the same and quite small under high normal stress. It can be considered that the interface roughness may be another reason for the evolution law of permeability during the seepage process, which was also found by Xie et al. [19].

#### 3.2.2. Shear-Seepage Process

The water output versus time curves under the normal stress of 100 kPa and 1600 kPa are plotted in Figure 10.

It can be observed that the water output of the clay–mortar specimen is larger than that of the clay–metal specimen during the same seepage time, indicating that the clay–mortar specimen has great permeability. As shown in Figure 10, seven shear periods are divided by seven shear displacement control points, and each seepage duration lasts more than 6 h. During each seepage process, due to the shear action, the original stable state of specimen is broken, and the water output fluctuates in the beginning. As the seepage processes, the water output tends to be stable and increases linearly with time, adhering to Darcy’s law. The slope of the straight line is the seepage rate of specimen under the current shear displacement control point.

In Figure 10a, the water output versus time curve of the clay–metal specimen (blue curve) is steep and the slope is large in the initial seepage period. As the shear and seepage processes, the curve gradually becomes flatter and the slope becomes smaller. In the end period, the slope increases slightly. This indicates that the overall permeability coefficient of specimen has the characteristics of a rapid decrease at first and then a slight increase at last. However, the curve of the clay–mortar specimen (black curve) also shows a similar trend from steepness to gentleness in the initial stage, but in the last stage of the shear-seepage process, the curve becomes steeper again obviously. This reflects that the overall permeability coefficient of the specimen decreases first and then increases dramatically under the normal stress of 100 kPa. The results show that the interface seepage is related to interface roughness.

In Figure 10b, the water output versus time curves of two clay–structure interfaces are basically similar, and the overall curves present an obvious linear relationship. The water output of the two specimens in the same time is also almost the same. This indicates that under high normal stress, the specimen permeability changes little as shear displacement increases, and the effect of interface roughness on the interfacial seepage is also weak. It is observed that the results under 1600 kPa are quite different from those under 100 kPa, which shows that the normal stress has a significant effect on the shear-seepage process.

In addition, during the entire seepage process, there is no obvious turbidity in the water output, and the seepage state remains stable. Under other test schemes, the shear-seepage process is similar, and there is also no seepage failure phenomenon under the shear-seepage process. Even when the normal stress is 1600 kPa and the hydraulic gradient reaches 1280, the specimen can still maintain integrity, the seepage process remains stable, and the water output keeps clear. The experimental phenomenon demonstrates seepage failure is not induced when the specimen is subjected to large shear deformation.

#### 3.2.3. Overall Permeability Coefficient after Shear Deformation

Relationships between overall permeability coefficient of clay–structure specimen and shear displacement under different normal stress are shown in Figure 11. The three-dimensional cylinder represents overall permeability coefficient value at each shear displacement control point, and the same shear displacement control point is marked with the same color. The main view projection curve shows the relationships between overall permeability coefficient and shear displacement under the same normal stress.

Taking the clay–mortar specimen (see Figure 11b) as an example, the overall permeability coefficient of specimen after shear deformation is analyzed. From the longitudinal view, the overall permeability coefficient decreases with the increase of normal stress under the same shear displacement control point (the same color cylinder). Especially when the normal stress increases from 100 kPa to 200 kPa, the value drops by an order of magnitude. This suggests that the normal stress could reduce specimen permeability. From the lateral view, the overall permeability coefficient exhibits different evolution laws under low normal stress and high normal stress. Under low normal stress (σ = 100, 200 kPa), the overall permeability coefficient of specimen decreases rapidly at the beginning of shear and reaches the minimum when the shear displacement equals 200% of the peak strength displacement value. Then as shear continues, the value appears to increase dramatically in reverse. This phenomenon is more obvious under the normal stress of 100 kPa. Finally, it remains stable after large shear deformation. The similar results were also shown in Lei et al. [25] test results under the condition of vertical shearing and seepage. He concluded that the permeability of the interface with impurities decreases with shear deformation increase at the beginning and then increases after a certain shear deformation decreases. However, under high normal stress, the overall permeability coefficient of specimen decreases continuously with the increase of shear displacement, and the value reduction is very small. When the shear displacement reaches 3~5 mm, the overall permeability coefficient tends to be stable, and the value is at the level of 10^−9^ cm/s. The present findings indicate that specimen permeability evolution process is related to the mechanical properties. Note that overall permeability coefficient after shear never exceeds initial permeability coefficient under all test schemes. The ratio of the stabilized permeability coefficient to the initial value ranges from 0.6 to 0.8. This suggests that shear deformation can reduce specimen permeability instead of raising it. This phenomenon also appeared in Deng et al. [33] results. In other words, the most dangerous situation occurs in the initial condition (without shear). It is of great significance to control the initial permeability of clay–structure specimen to ensure the permeability stability in engineering projects.

The clay–metal specimen results are similar to the clay–mortar specimen results, except that the reverse increase of overall permeability coefficient after large shear deformation under low normal stress is not obvious. The overall permeability coefficient of clay–metal specimen generally decreases with the increase of shear displacement and tends to stabilize rapidly. In addition, the overall permeability coefficient of the clay–metal specimen is smaller than that of the clay–mortar specimen at the same shear displacement control point under low normal stress, whereas the values are almost the same under high normal stress. This indicates that the interface roughness has an obvious influence on specimen permeability under low normal stress.

## 4. Discussion

The shear dilation features of the clay–structure interface can be used to explain the clay–structure specimen permeability evolution process. During the initial shearing stage, the cohesion force of the clay–structure interface increases continuously as shear displacement increases. At this time, the shear dilation of the clay–structure interface is weak. The clay particles’ movements make the clay soil denser, which causes the void ratio reduces and the overall permeability coefficient decreases rapidly. As shear processes, the interfacial strength reaches its peak and the shearing failure occurs at the clay–structure interface. The clay particles’ ploughing movement on the interface causes the loss of cohesive force and decreases the frictional force. Under low normal stress, because the cohesion potential energy and friction potential energy released by the clay soil exceed the restraint effect of the normal stress, the clay–structure interface shows obvious shear dilation, which results in the reverse increase of overall permeability coefficient after large shear deformation. Furthermore, since the restraint effect of the normal stress of 100 kPa is less than that of the normal stress of 200 kPa, it is also more obvious that specimen permeability decreases first and then increases with the increase of shear displacement under the normal stress of 100 kPa. In addition, due to the interface roughness, the smooth clay–metal interface releases less energy, and the rough clay–mortar interface releases more energy. The shear dilation is more likely to occur on the rougher clay–mortar interface, which results in a more obvious reverse increase of permeability after large shear deformation. However, under high normal stress, the whole shear process shows a shrinkage phenomenon because the normal stress reduces the number of compressed clay particles and hinders ploughing movement. The clay–structure specimen keeps being compressed and compacted during shearing, which results in a continuous decrease in specimen permeability.

Another important factor in the permeability evolution laws of clay–structure specimen is the interface roughness of structure rods. Figure 12 shows the schematic diagram of clay–metal and clay–mortar interface elements in different contact states during shear.

As shown in Figure 12a, the clay–metal interface is almost smooth and clay particles are uniformly distributed on the interface in consolidation state. As shear occurs, surface clay particles will slip on the interface (see red balls) and other clay particles will occur dislocation movement inside clay soil, which make the clay soil more compact. Hence, the void ratio of clay will be reduced, and the height of the interface element will be narrowed, which results in a decrease in specimen permeability. However, in Figure 12b, there are some obvious interface pores on the clay–mortar interface (see black areas). Some clay particles will fill part of interface pores during consolidation. When in the initial shearing stage, the extremely fine clay particles will further fill interface pores and the black areas will be reduced, making clay particles more closely attached to the mortar interface. As shear displacement increases, the interface pore filling process is gradually completed, and specimen permeability reaches a minimum. The results suggest that the evolution process of interface pores during shear directly affects the specimen permeability evolution. The permeability decreases when the interface pores are filled. After that, the effect of shear dilation on specimen permeability is dominant. The height of the clay–mortar interface element in the shear state is higher than that in the consolidation state, which results in a reverse increase of permeability under low normal stress. In summary, due to the existence of interface pores, the permeability coefficient of clay–mortar specimen is larger than that of the clay–metal specimen. The specimen permeability under low normal stress increases with interface roughness increasing, but the effect of interface roughness is weak under high normal stress.

## 5. Conclusions

We propose a shear-seepage test system to investigate the characteristics of clay–structure interface seepage perpendicular to the shear direction after shear deformation. A shear test and a shear-seepage test on clay–metal and clay–mortar interfaces are carried out. Based on the experimental results, the following conclusions can be drawn:Both the peak and residual shear strength of clay–structure interface increase as the normal stress increases. The shear strength and peak displacement values of clay soil, clay–mortar interface, and clay–metal interface decrease successively.The evolution laws of overall specimen permeability coefficient are different under low and high normal stress. The effect of interface roughness on the specimen permeability is remarkable under low normal stress.The overall permeability coefficient after shear deformation is less than its initial value under all test schemes. There is no seepage failure phenomenon on the clay–structure interface after large shear deformation in our tests.The evolution of both shear dilation and interface pore in the shear process controls clay–structure specimen permeability evolution with shear deformation.

## Figures and Tables

**Figure 1 materials-15-03802-f001:**
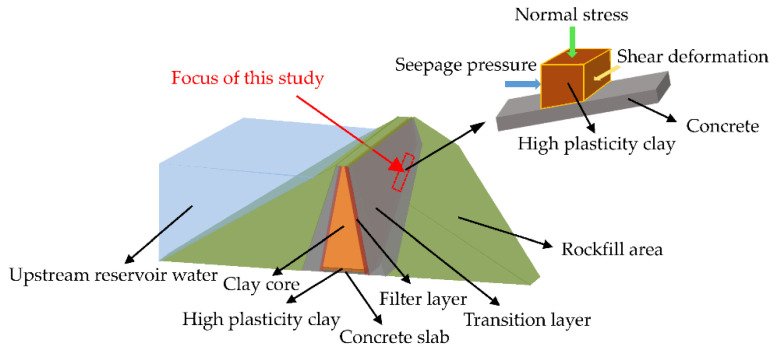
Working condition of clay–structure interface in high clay–core rockfill dams.

**Figure 2 materials-15-03802-f002:**
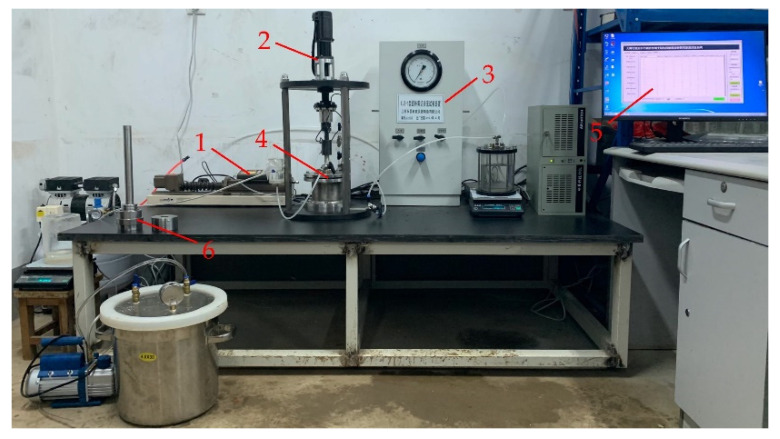
Image of the shear-seepage test system: 1 confining pressure system, 2 rotary shear system, 3 seepage pressure system, 4 shear-seepage pressure chamber, 5 automatic data acquisition system, 6 specimen preparation system.

**Figure 3 materials-15-03802-f003:**
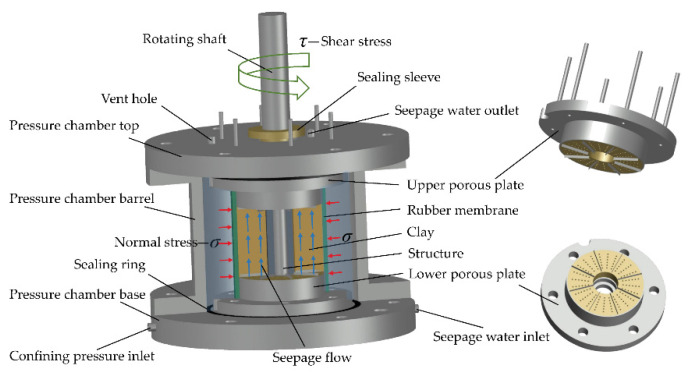
Three-dimensional schematic diagram of the shear-seepage pressure chamber.

**Figure 4 materials-15-03802-f004:**
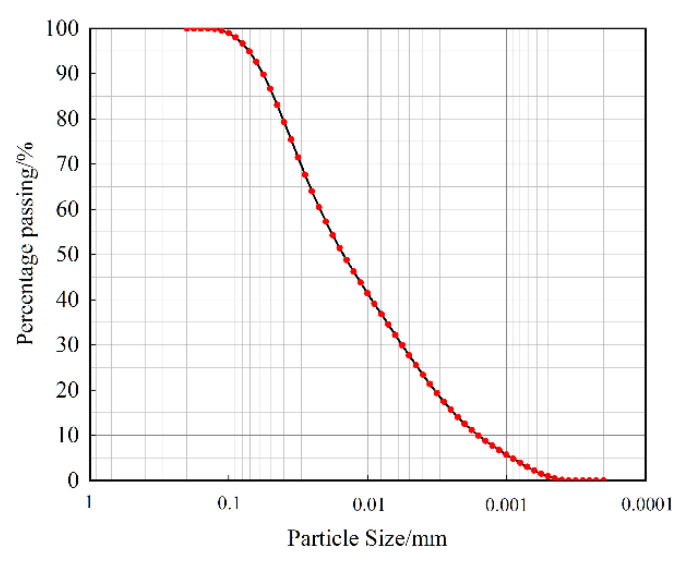
Grain size distribution curve of the clay soil.

**Figure 5 materials-15-03802-f005:**
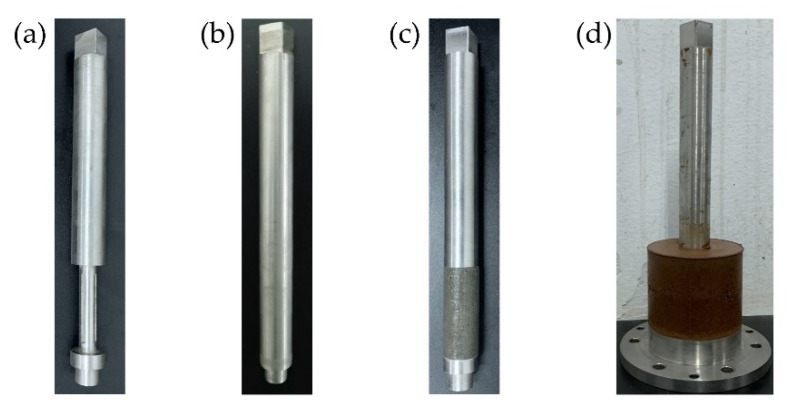
The structure rods and clay–structure specimen: (**a**) rotating shaft; (**b**) metal rod; (**c**) mortar rod; (**d**) clay–structure specimen.

**Figure 6 materials-15-03802-f006:**
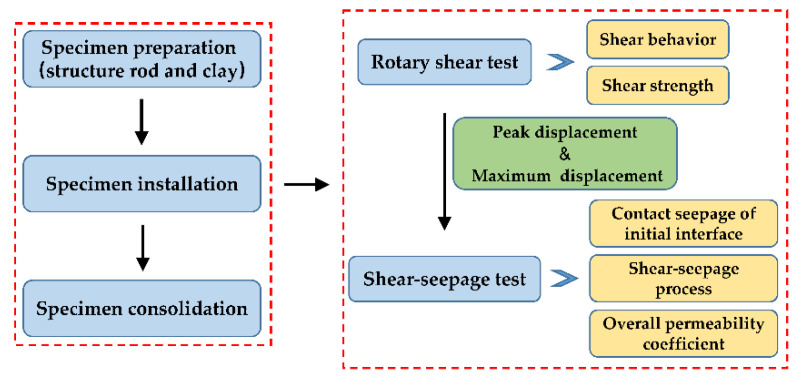
The experimental flowchart in this study.

**Figure 7 materials-15-03802-f007:**
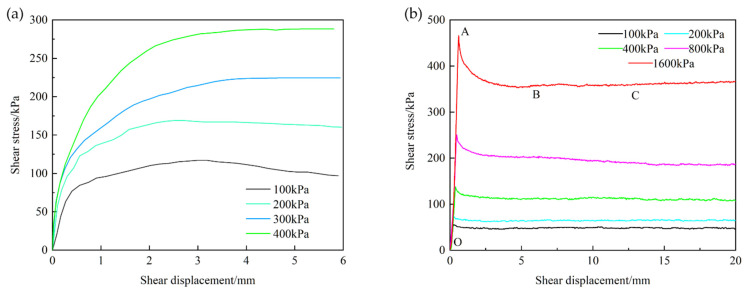
Relationships between shear stress and shear displacement: (**a**) clay soil; (**b**) clay–metal interface; (**c**) clay–mortar interface.

**Figure 8 materials-15-03802-f008:**
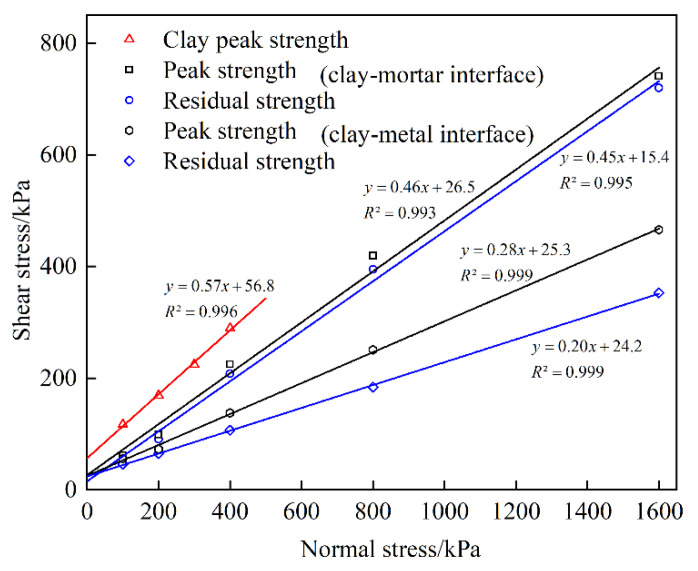
Relationships between normal stress and shear stress.

**Figure 9 materials-15-03802-f009:**
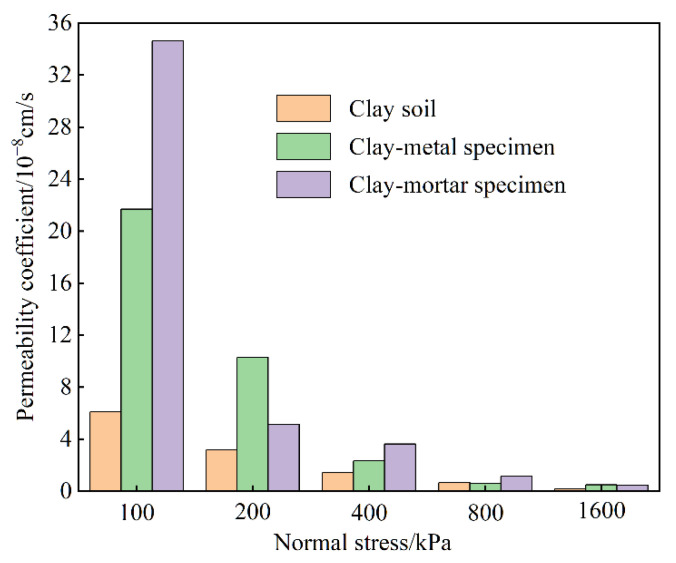
Contact seepage results of initial interface.

**Figure 10 materials-15-03802-f010:**
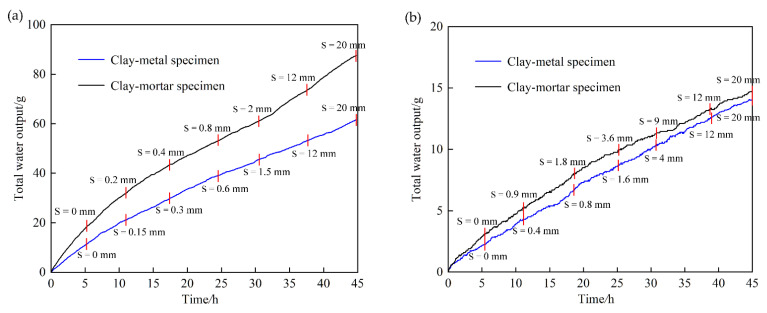
The water output versus time curves under different normal stresses: (**a**) 100 kPa; (**b**) 1600 kPa.

**Figure 11 materials-15-03802-f011:**
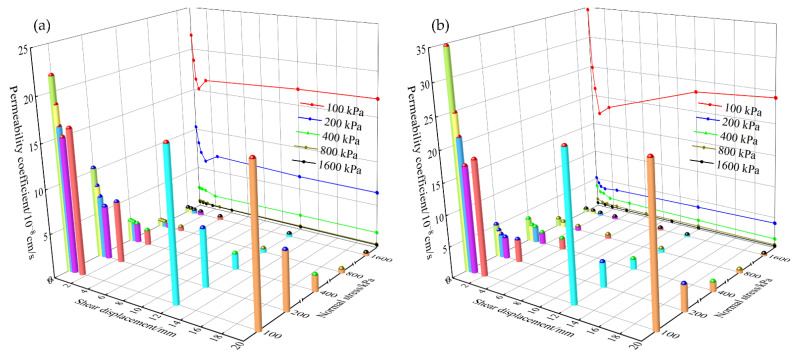
Relationships between overall permeability coefficient and shear displacement under different normal stress: (**a**) clay–metal specimen; (**b**) clay–mortar specimen.

**Figure 12 materials-15-03802-f012:**
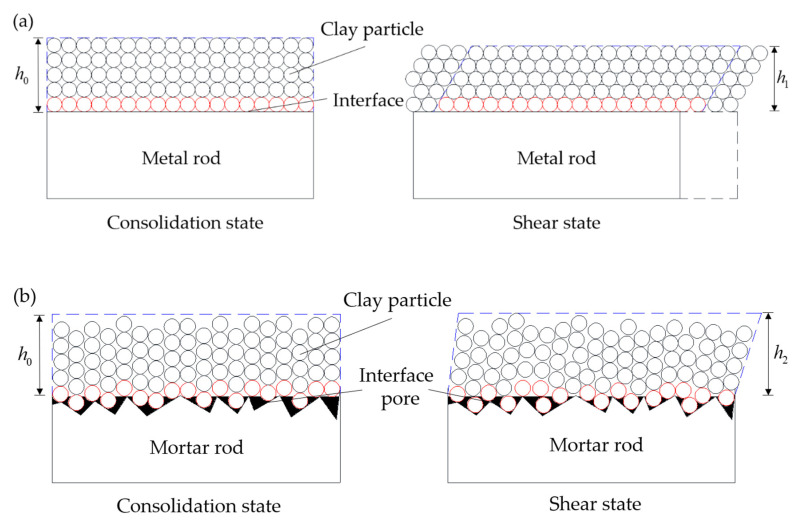
Schematic diagram of clay–structure interface: (**a**) metal rod; (**b**) mortar rod.

**Table 1 materials-15-03802-t001:** Basic physical properties of the clay soil.

Liquid Limit $\left(w_{L})\;(\%\right)$	Plastic Limit $\left(w_{p})\;(\%\right)$	Plasticity Index $\left(I_{p}\right)$	Specific Gravity $\left(G_{s}\right)$	Optimum Moisture Content $\left(w_{op})\;(\%\right)$	Maximum Dry Density $\left(\rho_{d\max})\;(g/{\mathrm{cm}}^{3}\right)$
31.0	15.7	15.3	2.725	15.6	1.83

**Table 2 materials-15-03802-t002:** Testing schemes.

Type	Normal Stress (kPa)	Shear Displacement Control Point (mm)	Seepage Pressure (kPa)	Structure Rod
Rotary shear test	100, 200, 400, 800, 1600	0~20	-	Metal rod
				Mortar rod
Shear-seepage test	100	0, 0.15, 0.3, 0.6, 1.5, 12, 20	40	Metal rod
	200	0, 0.25, 0.5, 1, 2.5, 12, 20	80	
	400	0, 0.2, 0.4, 0.8, 2, 12, 20	160	
	800	0, 0.25, 0.5, 1, 2.5, 12, 20	320	
	1600	0, 0.4, 0.8, 1.6, 4, 12, 20	640	
	100	0, 0.2, 0.4, 0.8, 2, 12, 20	40	Mortar rod
	200	0, 0.25, 0.5, 1, 2.5, 12, 20	80	
	400	0, 0.4, 0.8, 1.6, 4, 12, 20	160	
	800	0, 0.6, 1.2, 2.4, 6, 12, 20	320	
	1600	0, 0.9, 1.8, 3.6, 9, 12, 20	640	

**Table 3 materials-15-03802-t003:** Strength parameters of clay and clay–structure interfaces.

Material Tested	$c_{p}$	$\varphi_{p}$	$c_{r}$	$\varphi_{r}$
Clay soil	56.8	29.7	-	-
Clay–metal interface	25.3	15.6	24.2	11.3
Clay–mortar interface	26.5	24.7	15.4	24.2

## Data Availability

The data presented in this study are available on request from the corresponding author.

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
