# Peer review of "Experimental Investigation on Seepage Characteristics of Clay–Structure Interface after Shear Deformation"

_materials, 2022, doi:10.3390/ma15113802_

Round 1
Reviewer 1 Report
The paper "Experimental investigation on seepage characteristics of clay-structure interface after shear deformation" is well written by the authors and presented in a well-structured way, I consider it suitable for publication in Materials after minor corrections:
(1) The abstract may present more quantitative results and details of the experimental procedure performed by the authors;
(2) Although the objectives are clear to readers, there is a need for the authors to highlight the innovation of this research at the end of the introduction, what makes this paper different from others already published on the subject??
(3) An experimental flowchart can be added by the authors;
(4) "The hindering effect of normal stress is a reason for the specimens' ductility improvement under higher normal stress. The above peak displacement values ​​will be used as shear displacement control points in the subsequent shear-seepage tests." This sentence needs to be better explained by the authors!
(5) In the results section the authors need to explore more to compare their research with other results in the area, this is very weak in this section in general!
Reviewer 2 Report
This is a good paper that contains good sets of experiments what is missing is the application of these findings to other areas of research such as EOR, CO2 capture etc... examples are ' 1) Study of the effect of clay particles on low salinity water injection in sandstone reservoirs'
2) Insight into Enhancement of CO2 Captured by Clay Minerals
The other question is what kind of clay , more details of type of clay i needed and if these findins are applicable ?
Those could be part of introduction , and could be expanded more specifically with the measured properties.
In general these rest of the paper looks find and acceptable.
Reviewer 3 Report
The manuscript entitled "Experimental Investigation on Seepage Characteristics of Clay- 2 Structure Interface After Shear Deformation" by Tan et al., is an interesting paper. However, the following questions should be addressed for possible publication.
1. I am wondering if you consider the effects of scales, like scale upgrade from experiments to real dam. The experimental scale and the real dam scale are of great different. How you design the experimental parameters, like the shear stress, seepage pressure et al., depending on your devices? I think these should be designed according to the real dam circumstance. And what’s these parameters in real dam, please list some or give a range.
2. How the experimental results help with dam design, how your results help in real dam parameters design, could you give an example.
Reviewer 4 Report
Confidential Review Comments for materials-1729149
The article deals with studying the interface behaviour between clay core and concrete targeting seepage characteristics. The article is practical oriented and the reviewer lauds the efforts gone into taking up this need of the hour work. The manuscript is well written, analysis follow logical sequence, references are properly cited and conclusions are apt. The authors must address/clarify the following comments while revising their manuscript.
- Lines 100-116 – The authors must report the shear rotation rate adopted in the study as it will have direct bearing in quantifying the seepage characteristics.
- How does the aspect ratio adopted in the study relate to realistic field scenario and how does it affect the prevailing field seepage values? Will there be any underestimation or overestimation of seepage values?
- Labelling for Fig 3 can be better. The authors can provide the nomenclature (i.e., schematics) directly in the figure rather than providing them as part of figure caption. This figure can be redrawn while revising the manuscript.
- What precautions are taken to ensure that water doesn’t enter from sealed cavities apart from rubber membrane acting as barrier?
- What are the intrinsic and extrinsic factors affecting the specimen preparation system when the rotational speed mismatches with rotational shear-seepage test apparatus? (Line 132)
- Provide USCS Classification of the studied clay in Table 1.
- Maintain same tense in Section 2.3 (Follow Simple Past Tense and avoid Simple Present Tense while describing different testing stages)
- How do the results appear if the normal stress is kept constant and seepage pressure(s) are varied? (Refer Table 2).
- The equation 1 must be duly referenced relying on either ASTM code or standard text book.
- How does the inherent moisture content affect the strength parameters? (Refer Table 3)
- What are the typical normal stress values encountered in a 100 ft composite earthen dam? How does the current study capture the permeability coefficients for such a field scenario? Refer Fig 8.
- Provide water output versus time for a normal stress condition of 1600 kPa similar to the Fig 8, while revising the manuscript. Compare the results for 100 kPa and 1600 kPa and bring out salient features.
- The friction component plays a vital role in clay-metal and clay-mortar specimens and is well established. How do the authors quantify the results based on mortar roughness affected seepage behaviour? Relying on Fig 11, the authors must bring out some qualitative aspects.
- Rephrase conclusions 1 and 2 as most of the content seems repetitive from earlier sections.

Round 2
Reviewer 4 Report
The authors have addressed the comments raised by me during first phase of review.